# A Pattern of Care Report on the Management of Patients with Squamous Cell Carcinoma of the Anus—A Study by the Italian Association of Radiotherapy and Clinical Oncology (AIRO) Gastrointestinal Tumors Study Group

**DOI:** 10.3390/medicina57121342

**Published:** 2021-12-09

**Authors:** Pierfrancesco Franco, Giuditta Chiloiro, Giampaolo Montesi, Sabrina Montrone, Alessandra Arcelli, Tiziana Comito, Francesca Arcadipane, Luciana Caravatta, Gabriella Macchia, Marco Lupattelli, Marina Rita Niespolo, Fernando Munoz, Elisa Palazzari, Marco Krengli, Francesca Valvo, Maria Antonietta Gambacorta, Domenico Genovesi, Giovanna Mantello

**Affiliations:** 1Department of Translational Medicine, University of Eastern Piedmont, Via Solaroli 17, 28100 Novara, Italy; marco.krengli@med.uniupo.it; 2Department of Radiation Oncology, ‘Maggiore della Carita’ University Hospital, Via Solaroli 17, 28100 Novara, Italy; 3Fondazione Policlinico Universitario ‘Agostino Gemelli’ IRCCS, Via della Pineta Sacchetti, 00168 Rome, Italy; giuditta.chiloiro@policlinicogemelli.it (G.C.); mariaantonietta.gambacorta@policlinicogemelli.it (M.A.G.); 4Radiation Oncology Department, ‘S.M. Della Misericordia’ Hospital, AULSS 5 Veneto, Viale Tre Martiri 140, 45100 Rovigo, Italy; giampaolo.montesi@aulss5.veneto.it; 5Radiation Oncology Unit, Pisa University Hospital, Via Roma 67, 56123 Pisa, Italy; sabrina.montrone@ao-pisa-toscana.it; 6Radiation Oncology Unit, IRCCS Azienda Ospedaliero-Universitaria di Bologna, Via Massarenti 9, 40138 Bologna, Italy; alessandra.arcelli2@unibo.it; 7Department of Experimental, Diagnostic and Specialty Medicine-DIMES, Alma Mater Studiorum, Bologna University, Via Massarenti 9, 40138 Bologna, Italy; 8Radiotherapy Department, Humanitas Clinical and Research Hospital IRCCS, Via Manzoni 56, Rozzano, 20089 Milan, Italy; tiziana.comito@cancercenter.humanitas.it; 9Department of Oncology, Radiation Oncology, AOU ‘Citta’ della Salute e della Scienza’, Via Cavour 31, 10123 Turin, Italy; farcadipane@cittadellasalute.to.it; 10Radiation Oncology Unit, ‘SS Annunziata Hospital’, ‘G. D’Annunzio’ University of Chieti-Pescara, Via dei Vestini, 66100 Chieti, Italy; luciana.caravatta@asl2abruzzo.it (L.C.); d.genovesi@unich.it (D.G.); 11Radiation Oncology Unit, Gemelli Molise Hospital–Universita’ Cattolica del Sacro Cuore, Largo Agostino Gemelli 1, 86100 Campobasso, Italy; gabriella.macchia@gemellimolise.it; 12Radiation Oncology Section, Perugia General Hospital, Piazzale Meneghini 1, 06156 Perugia, Italy; marco.lupattelli@ospedale.perugia.it; 13Department of Radiation Oncology, Azienda Ospedaliera S. Gerardo, Via Pergolesi 33, 20900 Monza, Italy; r.niespolo@hsgerardo.org; 14Radiation Oncology Department, ‘Umberto Parini’ Regional Hospital, AUSL Valle d’Aosta, Aosta, Viale Ginevra 3, 11100 Aosta, Italy; FMunoz@ausl.vda.it; 15Radiation Oncology Department, Oncological Referral Center, Via Gallini 2, 33081 Aviano, Italy; elisa.palazzari@cro.it; 16Scientific Direction Unit, National Center for Oncological Hadrontherapy (CNAO), Strada Campeggi 53, 27100 Pavia, Italy; francesca.valvo@cnao.it; 17Department of Neuroscience, Imaging and Clinical Sciences, ‘G.D’Annunzio’, University of Chieti-Pescara, Via dei Vestini 31, 66100 Chieti, Italy; 18Department of Oncology and Radiotherapy, Azienda Ospedaliero Universitaria Ospedali Riuniti, Via Conca 71, Torrette, 60002 Ancona, Italy; giovanna.mantello@ospedaliriuniti.marche.it

**Keywords:** anal cancer, squamous cell carcinoma, anus, chemoradiation, radiotherapy, pattern of care

## Abstract

*Background and objectives:* The diagnosis and therapy of squamous cell carcinoma of the anus may vary significantly in daily clinical practice, even if international guidelines are available. *Materials and Methods:* We conducted a pattern of care survey to assess the management of patients with anal cancer in Italy (38 questions). We analyzed 58 questionnaires. *Results:* Most of the respondents work in public and/or university hospitals (75.8%) in northern Italy (65.5%). The majority (88.0%) treat less than 20 patients/year. Common examinations for diagnosis and staging are anorectal endoscopy (84.5%), computed tomography scan (86.2%) and pelvic magnetic resonance imaging (MRI) (96.5%). The most frequently prescribed dose to primary tumor is 50–54 Gy (46.5–58.6%) for early stage disease and 54–59.4 Gy (62.1–32.8%) for locally advanced cases. Elective volumes are prescribed around 45 Gy (94.8%). Most participants use volumetric intensity modulated radiotherapy (89.7%) and a simultaneous integrated boost (84.5%). Concurrent radiotherapy, 5-fluorouracil and mitomycin is considered the standard of care (70.6%). Capecitabine is less frequently used (34.4%). Induction chemotherapy is an option for extensive localized disease (65.5%). Consolidation chemotherapy is rarely used (18.9%). A response evaluation is conducted at 26–30 weeks (63.9%) with a pelvic MRI (91.4%). Follow-up is generally run by the multidisciplinary tumor board (62.1%). *Conclusions:* Differences were observed for radiotherapy dose prescription, calling for a consensus to harmonize treatment strategies.

## 1. Introduction

The standard therapeutic option for patients with non-metastatic squamous cell carcinoma (SCC) of the anus comprises of a combined modality treatment with concurrent chemoradiation based on 5-fluorouracil (5-FU) and mitomycin C (MMC) [1,2,3]. In the setting of advanced and/or metastatic disease, various chemotherapy regimens are available, leading to different outcome results and toxicity profiles [4,5]. Radiotherapy (RT) delivery is based on intensity modulated radiotherapy (IMRT), usually supported by image guidance (IGRT) [6,7]. Several national and international guidelines provide indications for diagnosis, staging, treatment and follow-up [8,9,10,11,12]. In 2014, the Italian Association for Radiotherapy and Clinical Oncology (AIRO), developed national guidelines dedicated to the RT treatment of tumors of the gastrointestinal tract, including SCC of the anus, to optimize and homogenize treatment approaches within the Italian radiation oncology community [13]. Nevertheless, a recent retrospective observational study performed by the Gastrointestinal Tumors Study Group of AIRO reported some degree of heterogeneity in terms of RT dose prescription, treatment volume selection and definition, delivery approaches and preference of different combination therapy modalities [14]. To provide a backbone for the update of the Italian national guidelines and to harmonize treatment recommendations for anal cancer in Italy, we ideated and conducted a national survey to shed light on how patients with SCC of the anal canal are currently diagnosed and treated in the country. We report on the results of the survey, framing them within the available evidence for the diagnosis and treatment of anal SCC.

## 2. Materials and Methods

The project was developed within the Study Group for Gastrointestinal Tumors of AIRO, whose Directive Council acted as a steering committee. An external panel of radiation oncologists with a specific expertise in the management of anal cancer provided suggestions and comments. Face validity, together with the content, wording and general flow of the survey was internally evaluated. An online cross-sectional survey was carried out using Survey Monkey (www.surveymonkey.com; accessed on 28 March 2021), with an automatic method for capturing responses. Usability and technical functionality of the electronic form was assessed before fielding the questionnaire. No personal information was collected. Professional information was stored within the Survey Monkey platform and protected from unauthorized access, as compliant with the platform regulatory. The project was approved by the Scientific Council and the Board of Directors of AIRO. Participants were invited to participate voluntarily (February–March 2021) via email, after identification as members of AIRO. The invitation was sent by the Secretariat of AIRO. One reminder was sent during the study period. No explicit informed consent was requested. No incentive was offered. The only two requirements to be eligible to participate in the survey were to work as a radiation oncologist and to be an active member of AIRO with an expertise in the treatment of gastrointestinal malignancies including anal cancer. The survey was set as a ‘closed survey’ with a selection of participants upon invitation. The initial contact mode to the participant was made via Internet. The initiative was announced and advertised through the network of AIRO using both its mailing list and website. The English translation of the exact wording used for the announcement can be found in the Appendix A. One radiation oncologist per center was allowed to participate in the survey. Demographics and professional information useful for stratification were collected. The questionnaire consisted of 38 questions, some of them allowing for multiple answers and comments, and covering diagnosis and treatment of SCC of the anal canal and margin (Appendix A). Statistical analysis was provided by www.surveymonkey.com (accessed on 28 March 2021) and included a description of all variables. Responses were tabulated, and the percentage values reported. The survey was compliant with the CHERRIES guidelines for reporting results of internet e-surveys [15].

## 3. Results

Among the 165 RT departments documented in Italy by AIRO who were invited (as per 2018), a total of 71 centers (43%) participated in the present survey. Among them, 58 participants (82%) fully completed the questionnaires and their answers were considered for the current analysis. Detailed characteristics of the participants and centers can be found in Table 1. Most of the respondents work in public and/or university hospitals (75.8%) in the northern part of the country (65.5%). The clinical experience of the participants was almost equally split between below (48.3%) and above (51.7) 10 years. The vast majority of the centers (88.0%) treats less than 20 anal cancer patients per year.

### 3.1. Diagnosis and Staging

See Table 2 for details. With respect to diagnosis and staging, the most commonly prescribed examinations are anorectal endoscopy (84.5%), contrast-enhanced computed tomography (CT) scan of the thorax and abdomen (86.2%) and pelvic magnetic resonance imaging (MRI) (96.5%). Pelvic MRI is considered as a mandatory examination for diagnosis and staging by most of the respondents (86.2%), while 18-fluorodeoxyglucose-positron emission tomography (FDG-PET) is mostly deemed as an optional or second-level examination (62.1%). Biopsy confirmation of a suspected inguinal lymph node is limited to cases with low or no metabolic uptake at functional imaging (60.3%). Screening for the human immunodeficiency virus is undertaken by half of the respondents (50.0%) by default, while determination of the human papilloma virus is consistently carried out by more than a half of the participants (58.6%). Multidisciplinary team discussion is considered standard by 87.9% of the centers.

### 3.2. Radiotherapy Dose Prescription and Delivery

See Table 3 for details. To properly define the primary gross tumor volume (GTV), most of the participants employ MRI (89.7%) and/or FDG-PET (77.6%). The most frequently prescribed dose to primary GTV is around 50 Gy (46.5%) and 54 Gy (58.6%) for T1–T2 tumors and around 54 Gy (62.1%) or up to 59.4 Gy (32.8%) for T3–T4 disease. Nodal disease was mostly prescribed around 50 Gy if sized below 3 cm (58.6%) or around 55 Gy (86.2%) if sized above. Elective volumes are mostly prescribed around 45 Gy (94.8%). Most of the participants use volumetric intensity modulated radiotherapy techniques (89.7%) and employ a simultaneous integrated boost to deliver extra doses to the primary tumor (84.5%).

### 3.3. Combination Therapy

See Table 4 for details. Most of the participants (70.6%) consider concurrent RT and 5FU-MMC as standard of care in anal cancer patients. Two cycles are normally administered (81.0%) with an MMC dose of 10 mg/m^2^. Capecitabine is considered standard of care by 34.4% of respondents, while cisplatin (CDDP) is mostly used in cases of clinical contraindication to MMC (70.7%). Induction chemotherapy is considered a viable option in cases of extensive localized disease (65.5%), mainly with 5FU-CDDP (56.9%). Consolidation chemotherapy is rarely used (18.9%), mostly with 5FU-CDDP (6.9%). The most commonly prescribed chemotherapy regimen for advanced or metastatic disease is CDDP-5FU (62%). HIV-positive patients are treated with standard concurrent chemoradiation in cases of normal CD4 positive count (39.6%), eventually requiring undetectable viral RNA (20.7%).

### 3.4. Response Assessment, Salvage Therapies and Follow-Up

See Table 5 for details. Most of the participants perform a response evaluation at 26–30 weeks (63.9%), with a pelvic MRI (91.4%) and/or FDG-PET (58.6%). Biopsy assessment is only performed in cases of suspected residual disease (53.4%). Surgery is considered as a salvage curative option for persistent/recurrent disease by 93.0% of respondents, after discussion within the tumor board (62.1%). Follow-up and survivorship is managed by the radiation oncologist in 60.3% of the cases. The follow-up protocol is generally (60.3%) more intense during the first 2 years after treatment (every 3 months) and less frequent in the following 3 years (every 6 months).

## 4. Discussion

This is the first survey exploring the pattern of care in Italy for the diagnosis and treatment of patients affected with SCC of the anus. With respect to pre-treatment and staging imaging modalities, thorax and abdominal CT, together with pelvic MRI, are prescribed by the vast majority of respondents (>85%), while FDG-PET is used by two out of three participants. Pelvic MRI is considered a mandatory examination to be requested, in agreement with the updated European Society for Medical Oncology (ESMO) guidelines [8]. Conversely, FDG-PET is considered an optional or second-level examination by more than 60% of respondents. This is in line with the results of a recent survey carried out in German-speaking countries [16,17]. In the recent ESMO guidelines, FDG-PET is considered as an exam to be recommended, but not mandated [8]. It has to be noted that, apart from staging purposes, FDG-PET may be clinically useful to confirm or not suspicious features detected on MRI, to drive target volume selection and delineation and to define organs at risk for tailored IMRT approaches [18,19,20]. Interestingly, gynecological examination is performed by around one out of five participants, even if it is mandated in the ESMO guidelines during the diagnostic work-up [8]. The attitude toward the diagnostic biopsy of suspicious inguinal lymph nodes is rather cautious, with one out of three respondents never performing it and half of them requesting it in case of difficulties in the interpretation of the findings coming from morphologic and/or functional imaging. The latter approach is in line with the ESMO guidelines, suggesting a further characterization of enlarged inguinal nodes when confirmatory features of malignancy are lacking on either pelvic MRI or FDG/PET [8]. Routine HIV screening is performed by half of the centers participating in the survey, while around 40% would do it occasionally or in individuals at risk. The ESMO guidelines recommend testing in any individual whose lifestyle puts him/her at risk of contracting HIV infection, while NCCN guidelines suggest routine screening for HIV, since HIV-positive patients treated with highly active antiretroviral therapy may have similar treatment outcomes compared to HIV-negative patients [8,11,21]. The determination of HPV, as measured directly or by means of the overexpression of the surrogate marker p16, is performed by more than 90% of the participants. This may be important in terms of prognostic stratification, since HPV negative tumors are less likely to respond to definitive treatments, while patients with HPV/p16 positive tumors have improved survival [8,22]. The use of IMRT techniques has been declared by all the participants, similarly to what has been reported in the German-speaking country survey and suggested by NCCN and European guidelines [8,9,11]. The use of simultaneous integrated boost (SIB) techniques to the primary tumor and macroscopic lymph nodes to deliver extra doses is common in our survey (84.5%), even if almost half of respondents tend to employ also a sequential approach. Even in the absence of randomized data supporting the use of SIB, the RTOG 0529 trial, together with other mono-institutional series, suggest a trend towards a mild toxicity profile for IMRT and comparable oncological outcomes for SIB compared to a sequential boost [7,23,24,25]. Alternative boost techniques such as brachytherapy or contact therapy are used in few institutions. In our survey, we observed a rather large variability in terms of RT dose prescription, which reflects the corresponding heterogeneity present in the different national and international guidelines [8,9,10,11,12]. The most frequently prescribed total doses ranged between 50–55 Gy for early stage primary tumors and 54–59.4 Gy for advanced primaries. For involved lymph nodes, participants declared to prescribe doses around 50 Gy for small sized nodes and 54–56 Gy for larger nodes. The dose to elective nodal volumes is mostly around 45 Gy. Differences in terms of prescription doses to the primary tumor do exist, ranging from the conventionally fractionated 50.4 Gy used in the ACT II trial in the UK, to the 55–59 Gy employed in the RTOG 9811 trial for locally advanced disease, and the 60 Gy reached within observational studies in Northern European countries [1,2,26]. Dose escalation was specifically evaluated within the ACCORD 03 prospective phase III trial, which showed no benefit for dose escalation beyond 60 Gy, particularly when radiation dose is delivered with a time gap between the first phase of treatment and the sequential boost [27]. However, a recent pooled analysis of patient data enrolled in the ACCORD 03 and the KANAL phase 2 trials reported a total dose >60 Gy to be associated with better colostomy free survival [28]. The ongoing PLATO umbrella trial is currently assessing the efficacy and toxicity profile of risk-adapted RT dose prescription in anal cancer based on clinical staging [29]. In particular, the ACT4 study is a randomized phase II trial, targeting patients with early stage disease (T1, T2 up to 4 cm, N0) and comparing standard chemoradiation (50.4 Gy to the tumor, 40 Gy to the elective nodal region in 28 fractions + 5FU or capecitabine/MMC) with a reduced dose regimen (41.4 Gy and 34.5 Gy in 23 fractions, to primary tumor and elective volumes, respectively). The ACT5 trial targets patients with locally advanced tumors (T2N1–3 or T3/4Nany) and compared standard chemoradiation (53.2 Gy in 28 fractions + 5FU or capecitabine/MMC) with escalated regimens delivering a dose of either 58.8 or 61.6 Gy in 28 fractions. The aforementioned studies will help establish a risk-adapted radiotherapy dose prescription strategy in anal cancer. More than 90% of respondents usually deliver around 45 Gy to the elective volumes, which is in line with the recommendations of most guidelines [8,9,10,11,12]. Nevertheless, it has to be noted that the total dose prescribed in the ACT II trial to elective volumes was 30.6 Gy, given with conventional fractionation, and that a recent report from the UK highlighted the oncological safety of delivering low total dose with low dose per fraction during SIB-based image-guided RT in this setting (40 Gy in 28 fractions; 1.43 Gy/daily) [1,30]. Concurrent RT and two cycles of 5FU-MMC is considered standard by most respondents, as in most of the international guidelines [8,9,11,12]. The role of the second infusion of MMC has been debated since it may not provide benefit in terms of oncologic outcomes, conversely adding extra adverse events, particularly in terms of hematologic toxicity [31,32]. The use of capecitabine is considered standard by one third of the respondents, which reflects the low evidence of the available clinical data mostly relying on relatively small series [8]. Nevertheless, the results of the UK national cohort seem reassuring with respect to the general oncological safety of the use of capecitabine [33]. Cisplatin is used concurrent to RT only in cases of contraindication to MMC by most respondents, as suggested in the national and international guidelines [8,9,10,11,12]. Induction and consolidation chemotherapy are not considered standard, even if two out of three of the participants may consider primary systemic therapy in cases of very advanced loco-regional disease. This probably comes with the knowledge that some cases with extra-pelvic disease may be converted to cure [34]. Most of the participants use the doublet 5FU-CDDP as first-line therapy for advanced/metastatic cases, while up to one third employ carboplatin and paclitaxel, which is now considered standard after the publication of the InterAAct trial [4]. Almost one third of respondents would perform a response assessment after definitive treatment at 26 weeks upon initiation of concurrent chemoradiation, approximately 5 months upon its completion. This is in line with the results of the ACT II trial [35]. Interestingly, another third of respondents would wait until 6 months from treatment end to assess response. This probably reflects the low response kinetics of some tumors, to be taken advantage of, particularly in the case of a good response trajectory. Pelvic MRI is the most commonly used imaging modality for response assessment, as recommended in the recent ESMO guidelines [4]. Bioptic confirmation is undertaken only in cases of the suspicion of persistent/recurrent disease. The most common approach to treat recurrent/persistent local disease is surgery, which, as recommended, should include an extralevator abdominoperineal excision [4]. The clinical follow-up strategy of treated patients is reported to be in line with the guideline recommendations, suggesting an evaluation every 3–6 months for a period of two years, and every 6–12 months until five years. During follow-up, it is crucial to evaluate and take care of eventual conditions affecting the anorectal and sexual function, together with urinary continence [4]. Treatment decisions, follow-up examinations and interventions are taken by the radiation oncologist, relying on a shared decision-making process taken within the tumor board, which is considered a fundamental instrument to improve patient care in this setting [36].

## 5. Conclusions

The present study has some limitations. Sampling and non-response biases could be present, due to the response rate of 43% and the unknown characteristics of those who did not respond, which did not allow us to control for the decreased chance for some of the potential respondents to be surveyed and for a selected response trend for certain participants. Order bias could also be present, with an influence of the format employed on the chance to provide a specific response. The study relied on self-reporting, known to be potentially misaligned with reality and leading to potential recall and response biases. In this sense, performance-based instruments or structured interviews, less influenced on the individuals’ awareness, are considered more reliable options in addition to traditional self-reported measures.

Nevertheless, a positive degree of concordance with national and international guidelines is reported in the present survey amongst radiation oncologists treating anal cancer patients in Italy. However, several differences have been observed in terms of RT prescription doses, HIV screening and the involvement of gynecologists in the initial patient’s assessment. These data underline the need for a consensus to further harmonize the management of anal cancer patients in the country. The Study Group for Gastrointestinal Tumors at AIRO is presently planning to set up a project based on the Delphi consensus methodology, to homogenize diagnosis and treatment for anal cancer patients in Italy in order to mitigate the differences in patient management outlined by the present survey. After the generation of a consensus, the Italian radiation oncology community is also planning to register all anal cancer cases within a shared platform in the frame of a prospective observational cohort study.

## Figures and Tables

**Table 1 medicina-57-01342-t001:** Characteristics of the participants and centers.

Radiotherapy Facility	N (%)
Public	30 (51.7)
Accredited private hospital	7 (12.1)
University Hospital	14 (24.1)
Accredited cancer center (IRCCS)	7 (12.1)
**Operating region in Italy**	
Northern Italy	38 (65.5)
Central Italy	13 (22.4)
Southern Italy	7 (12.1)
**Years of experience in RT**	
<5	10 (17.2)
5–10	18 (31.1)
11–15	9 (15.5)
>15	21 (36.2)
**Anal cancer patients treated/year**	
<10	23 (39.7)
11–20	28 (48.3)
21–30	6 (10.3)
>30	1 (1.7)
**MDT dedicated to anal cancer**	
Yes	54 (93.1)
No	4 (6.9)

Legend: N: number; IRCCS: Istituto di Ricovero e Cura a carattere scientifico; RT: radiotherapy; MDT: Multidisciplinary Team.

**Table 2 medicina-57-01342-t002:** Diagnosis and staging.

Staging Examinations (Multiple Answers Allowed)	N (%)
Rigid anal-rectal endoscopy	49 (84.5)
Colonoscopy	30 (51.7)
GYN evaluation + colposcopy	13 (22.4)
Contrast-enhanced CT scan (thorax-abdomen)	50 (86.2)
Pelvic MRI	56 (96.5)
Whole-body ^18^FDG-PET	39 (67.2)
Endoscopic ultrasound	19 (32.8)
**Attitude towards pelvic MRI at diagnosis**	
Mandatory	50 (86.2)
Optional but useful	6 (10.3)
Second-level examination	2 (3.5)
Useless	0 (0)
**Attitude towards ^18^FDG-PET at diagnosis**	
Mandatory	22 (37.9)
Optional but useful	20 (34.5)
Second-level examination	16 (27.6)
Useless	0 (0)
**Inguinal biopsy/fine needle aspiration of suspicious node**	
Always	3 (5.2)
Only if clinically palpable lymph node detected on CT (size > 1 cm) and ^18^FDG-PET avidity	1 (1.7)
Only in case of clinically palpable lymph node detected on CT (size > 1 cm) and borderline ^18^FDG-PET avidity	30 (51.7)
Only in case of clinically palpable lymph node detected on CT (size > 1 cm) without ^18^FDG-PET avidity	5 (8.6)
Never	19 (32.8)
**HIV screening (on blood or saliva)**	
Always	29 (50.0)
Sometimes	16 (27.6)
Only in case of risk factors	7 (12.1)
Never	6 (10.3)
**(HPV) p16 IHC detection on biopsy specimen**	
Always	34 (58.6)
Sometimes	21 (36.2)
Only in young patients	0 (0)
Only in clinical trials	3 (5.2)
Never	0 (0)
**Role of the multidisciplinary team**	
Standard approach for all patients	51 (87.9)
Necessary only in selected cases	5 (8.6)
Not applicable to my clinical practice	2 (3.5)

Legend: N: number; GYN: gynecological; CT: computed tomography; MRI: magnetic resonance imaging; FDG-PET: fluorodeoxyglucose positron emission tomography; HIV: human immunodeficiency virus; HPV: human papilloma virus; IHC: immunohistochemistry.

**Table 3 medicina-57-01342-t003:** Radiotherapy dose prescription and delivery.

Imaging for GTV Definition (Both Primary Tumor and Lymph Nodes) (Multiple Answers Allowed)	N (%)
Planning CT	8 (13.8)
Pelvic CT	19 (32.8)
Pelvic MRI	52 (89.7)
^18^FDG-PET	45 (77.6)
**RT delivery technique (multiple answers allowed)**	
3DCRT	0 (0)
IMRT	10 (17.2)
Volumetric IMRT	52 (89.7)
Tomotherapy	12 (20.7)
MRgRT	0 (0)
**Primary tumor boost (multiple answers allowed)**	
EBRT-Sequential boost	26 (44.8)
EBRT-SIB	49 (84.5)
EBRT-Electrons	2 (3.4)
Endocavitary or Contact BRT	3 (5.2)
Interstitial BRT	4 (6.9)
**Treatment after local excision for T1N0 tumor with risk factors**	
Exclusive RT with definitive dose	21 (36.2)
RT-CHT with RT dose de-escalation	13 (22.4)
RT-CHT with definitive RT dose	17 (29.3)
RT with dose de-escalation	2 (3.5)
Others	5 (8.6)
**RT dose to primary tumor GTV for T1–T2 tumors (dose range) (multiple answers allowed)**	
45–45.9 Gy	2 (3.5)
50–50.4 Gy	27 (46.5)
54–55 Gy	34 (58.6)
56–59.4 Gy	7 (12.1)
≥60 Gy	4 (6.9)
**RT dose to primary tumor GTV for T3–T4 tumors (dose range) (multiple answers allowed)**	
53 Gy	1 (1.7)
54–55-5 Gy	36 (62.1)
56–59.4 Gy	19 (32.8)
≥60 Gy	13 (22.4)
**Dose to elective volumes (multiple answers allowed)**	
30.6 Gy	1 (1.7)
36–37.5 Gy	2 (3.5)
42–42.5 Gy	5 (8.6)
45–45.9 Gy	55 (94.8)
49.5–50.4 Gy	11 (18.9)
>54 Gy	3 (5.2)
**Dose to involved nodes (sized < 3 cm) (multiple answers allowed)**	
40 Gy	1 (1.7)
45 Gy	1 (1.7)
50–51 Gy	34 (58.6)
52–53.2 Gy	6 (10.3)
54–56 Gy	19 (32.8)
59–59.4 Gy	2 (3.5)
≥ 60 Gy	4 (6.9)
**Dose to involved nodes (sized > 3 cm) (multiple answers allowed)**	
45 Gy	1 (1.7)
50–50.4 Gy	4 (6.9)
52–52.5 Gy	2 (3.5)
54–56 Gy	50 (86.2)
59–59.4 Gy	3 (5.2)
≥60 Gy	5 (8.6)

Legend: N: number; GTV: gross tumor volume; RT: radiotherapy; CT: computed tomography; MRI: magnetic resonance imaging; ^18^FDG-PET: fluorodeoxyglucose positron emission tomography; 3DCRT: 3-dimensional conformal radiotherapy; IMRT: intensity modulated radiotherapy; MRgRT: magnetic resonance guided radiotherapy; EBRT: external beam radiotherapy; SIB: simultaneous integrated boost; BRT: brachytherapy; CHT: chemotherapy.

**Table 4 medicina-57-01342-t004:** Combined modality treatment.

CHT Regimens Concurrent to RT	N (%)
5FU-MMC	41 (70.6)
5FU-CDDP	3 (5.2)
Cape-MMC	11 (19.0)
Cape-CDDP	1 (1.7)
Others	2 (3.5)
**Number of MMC cycles in cases of 5FU-MMC or Cape-MMC**	
1 cycle (week 1 of RT)	9 (15.5)
2 cycles (week 5–6 of RT)	47 (81.0)
Other	2 (3.5)
**MMC dose in cases of 5FU-MMC or Cape-MMC (1 MMC cycle)**	
10 mg/m^2^	21 (80.7)
12 mg/m^2^	2 (7.8)
10–12 mg/m^2^	3 (11.5)
**MMC dose in cases of 5FU-MMC or Cape-MMC (2 MMC cycle)**	
10 mg/m^2^	31 (91.2)
12 mg/m^2^	1 (2.9)
10–12 mg/m^2^	2 (5.9)
**Screening for DPYD genotype**	
Yes	46 (79.3)
No	12 (20.7)
**Use of Cape concurrent to MMC or CDDP and RT**	
Standard of care (daily practice)	20 (34.4)
Investigational (within clinical trial only)	4 (6.9)
Upon patient’s preference or in case of challenges for CVC placement	32 (55.2)
Other	2 (3.5)
**Use of CDDP as alternative to MMC concurrent to 5FU or Cape and RT**	
Equivalent to MMC	8 (13.8)
Inferior to MMC	8 (13.8)
Only in case of clinical contraindication to MMC	41 (70.7)
Other	1 (1.7)
**Use of induction chemotherapy**	
Standard	1 (1.7)
Not standard	19 (32.8)
Only in case of extensive pelvic involvement or extra-pelvic disease	38 (65.5)
Other	0 (0)
**Use of consolidation CHT after RT-CHT**	
Standard	0 (0)
Not standard	44 (75.9)
In case of high-risk disease (locally advanced tumors with nodal involvement)	11 (18.9)
Other	3 (5.2)
**CHT regimen for induction CHT**	
5FU-CDDP	33 (56.9)
5FU-MMC	17 (29.3)
Other	4 (6.9)
None	4 (6.9)
**CHT regimen for consolidation CHT**	
5FU-CDDP	4 (6.9)
5FU-MMC	3 (5.2)
Other	4 (6.9)
None	47 (81.0)
**Type of definitive RT-CHT in HIV+ve patients submitted to HAART**	
Standard CHT-RT	12 (20.7)
Standard CHT-RT in patient with normal CD4+ve count	23 (39.6)
Standard CHT-RT in patient with normal CD4+ve count and undetectable viral RNA	12 (20.7)
CHT dose reduction	4 (6.9)
Use of alternative CHT regimens (i.e., CDDP over MMC)	7 (12.1)
**Standard first-line chemotherapy for advanced or metastatic disease**	
CDDP-5FU	36 (62.0)
CBDCA + paclitaxel	19 (32.8)
(Modified) Docetaxel + CDDP + 5FU	3 (5.2)

Legend: N: number; CHT: chemotherapy; RT: radiotherapy; 5FU: 5-fluorouracil; Cape: capecitabine; MMC: mitomycin C; CDDP: cisplatin; Mg: milligrams; M^2^: square meters; DPYD: dihydropyrimidine dehydrogenase; CVC: central venous catheter; HIV: human immunodeficiency virus; +ve: positive; HAART: highly active antiretroviral therapy; CD4: cluster of differentiation 4; RNA: ribonucleic acid; CBDCA: carboplatin.

**Table 5 medicina-57-01342-t005:** Response assessment, salvage therapies and follow-up.

Optimal Timing for Restaging After the End RT-CHT	N (%)
8 weeks	6 (10.3)
3 months	10 (17.2)
6 months	20 (34.6)
>6 months	5 (8.6)
26 weeks	17 (29.3)
**Imaging examination for restaging after RT-CHT** (multiple answers allowed)	
Abdomino–pelvic contrast-enhanced CT scan	26 (44.8)
Pelvic contrast-enhanced MRI	53 (91.4)
^18^FDG PET-CT	34 (58.6)
Abdominal US	20 (34.5)
**Bioptic evaluation for response assessment**	
Always	2 (3.5)
Only if persistent disease is suspected or a residual scar is present	16 (27.6)
Only if persistent disease is suspected	31 (53.4)
I decide according to tumor clearance during RT-CHT	9 (15.5)
Never	0 (0)
**Opinion about salvage surgery for recurrent/persistent disease**	
Always curative	14 (24.1)
Curative in about half of patients	7 (12.1)
Never curative	0 (0)
My opinion is normally validated by tumor board	36 (62.1)
Other	1 (1.7)
**Treatment for local relapse**	
Exclusive surgery when feasible	54 (93.0)
Re-irradiation + CHT with palliative intent	0 (0)
Exclusive CHT	2 (3.5)
Re-irradiation + pre-operative CHT + eventual surgery	2 (3.5)
**Management of late toxicity in long-term survivors**	
Conducted by the radiation oncologist	35 (60.3)
Conducted by other specialists (medical oncologist, surgeon)	2 (3.5)
Based on tumor board management	21 (36.2)
**Follow-up timing**	
Every 3 months for the first 5 years	1 (1.7)
Every 6 months for the first 5 years	2 (3.5)
Every 3 months for the first year then every 6 months for the next 4 years	18 (31.0)
Every 3 months for the first 2 years then every 6 months for the next 3 years	35 (60.3)
Other	2 (3.5)

Legend: N: number; RT: radiotherapy; CHT: chemotherapy; CT: computed tomography; MRI: magnetic resonance imaging; FDG-PET: fluorodeoxyglucose positron emission tomography; US: ultrasounds.

## Data Availability

Data are available upon request to the corresponding author.

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
