# Peer review of "A Pattern of Care Report on the Management of Patients with Squamous Cell Carcinoma of the Anus—A Study by the Italian Association of Radiotherapy and Clinical Oncology (AIRO) Gastrointestinal Tumors Study Group"

_medicina, 2021, doi:10.3390/medicina57121342_

Round 1
Reviewer 1 Report
This is an interesting article regarding the management of patients with squamous cell carcinoma of the anus
I have the following comments:
- On the basis of what was the survey developed? Guidelines? Expert Opinion?
- Was there a steering committee?
- How many colleagues have been invited?
-
Who sent the questionnaires?
- Have any reminders been issued?
- Have you collected the gender of the participants?
- Why were the CHERRIES guidelines not used?
- Limitations of the study were not developed (recall bias etc.)
Author Response
Reviewer 1
This is an interesting article regarding the management of patients with squamous cell carcinoma of the anus.
We would like to thank the reviewer for the appreciation on our manuscript.
I have the following comments:
- On the basis of what was the survey developed? Guidelines? Expert Opinion?
The survey was developed on the basis of a Expert Opinion. This was added in the text within material and methods.
- Was there a steering committee?
The steering committee was represented by the Directive Council of the Study Group for Gastrointestinal Tumors of the Italian Association of Radiotherapy and Clinical Oncology (AIRO). This was added in the text within material and methods.
- How many colleagues have been invited?
A total of 165 departments were invited. This was better stated in the text.
- Who sent the questionnaires?
The invitation including the link to have access to the questionnaire was sent by the Secreteriat of the Italian Association of Radiotherapy and Clinical Oncology (AIRO). This was added in the main text.
- Have any reminders been issued?
One reminder was sent. This was added in the main text.
- Have you collected the gender of the participants?
Unfortunately, gender was not collected. We apologies for that.
- Why were the CHERRIES guidelines not used?
Thanks for the comment. We did not use them beacuse we were not aware of their exsistence. However, we followed the checklist proposed within the guidelines and stated when and how our survey was compliant with the suggestions. Thank you for the very helpful suggestion.
- Limitations of the study were not developed (recall bias etc.)
We addressed the limitations of the study, including sampling bias, non-response bias, response bias, order bias and recall bias. Thanks for this useful comment.
Reviewer 2 Report
The manuscript reports on a pattern-of-care survey with 38 questions about the management of patients with anal cancer in Italy and suggests the need for a consensus to further harmonise the management of anal cancer patients in the country. I have no comments and endorse publication of this manuscript in Medicina.
Strengths: The authors designed a detailed questionnaire with some alternative clinical measures and gave a full discussion about the results, especially in the diagnosis and treatment of anal cancer.Limitations: The manuscript did not mention how many participants were invited.
Author Response
Reviewer 2
The manuscript reports on a pattern-of-care survey with 38 questions about the management of patients with anal cancer in Italy and suggests the need for a consensus to further harmonise the management of anal cancer patients in the country. I have no comments and endorse publication of this manuscript in Medicina.
We would like to thank the reviewer for the appreciation on our manuscript.
Strengths: The authors designed a detailed questionnaire with some alternative clinical measures and gave a full discussion about the results, especially in the diagnosis and treatment of anal cancer.
Limitations: The manuscript did not mention how many participants were invited.
A total of 165 departments was invited. This was better stated in the text.
Reviewer 3 Report
It is well written and organized study. A statistical check is required. Native English check is recommended.
Author Response
Reviewer 3
It is well written and organized study. A statistical check is required. Native English check is recommended.
We would like to thank the reviewer for the appreciation on our manuscript.
Descriptive statistics has been double-checked and english language revised.
Reviewer 4 Report
I think it is appropriate to publish.
Author Response
Reviewer 4
I think it is appropriate to publish.
We would like to thank the reviewer for the appreciation on our manuscript.
Round 2
Reviewer 1 Report
Well done. Just a moderate english-improvement needed
This manuscript is a resubmission of an earlier submission. The following is a list of the peer review reports and author responses from that submission.
Round 1
Reviewer 1 Report
The concept of personalized cancer therapy has become very important in this decade, particularly for patients suffering from gastrointestinal cancer. The manuscript by Franco and coworkers analyze a pattern-of-care for the patients with squamous cell carcinoma of the anus treated with CCRT. This study could be helpful to develop the system to predict the survival of the patients with squamous cell carcinoma of the anus and can assist the physicians when assigning the therapeutic strategy to individual patients in Italy.
Comments:
- Although the references are classic, some of them are slightly outdated. It is recommended to supplement references related to your research area in the past five years, especially the high impact factor articles for the past three years.
- This manuscript is not well written in English and proof roughly. The context throughout the article should be corrected by a native-speaking English person.
Author Response
Reviewer 1.
The concept of personalized cancer therapy has become very important in this decade, particularly for patients suffering from gastrointestinal cancer. The manuscript by Franco and coworkers analyze a pattern-of-care for the patients with squamous cell carcinoma of the anus treated with CCRT. This study could be helpful to develop the system to predict the survival of the patients with squamous cell carcinoma of the anus and can assist the physicians when assigning the therapeutic strategy to individual patients in Italy.
Thanks for your comments and appreciation
Comments:
- Although the references are classic, some of them are slightly outdated. It is recommended to supplement references related to your research area in the past five years, especially the high impact factor articles for the past three years.
Thanks for pointing this out. Some of the references have been updated. Now 19/31 references were published in the last 5 years.

Reviewer 2 Report
The authors present a pattern of care report on the management of patients with squamous cell carcinoma of the anus. The questionnaire is comprehensive and well-constructed. The issues are of great importance for radiation oncologists. The authors provide a detailed discussion of recent literature and recent studies. There are no further remarks.
Second report(9.20):
The authors present a pattern of care report on the management of patients with squamous cell carcinoma of the anus. This topic is of great relevance for clinicians, since there is certainly a considerable heterogeneity in, e.g., staging examinations, radiotherapy dose prescription, and application of chemotherapy. Additionally, anal cancer is a rare tumor entity and radical treatment is often associated with high-grade toxicities. This poses a challenge to harmonize treatment strategies for optimal outcomes and to avoid unnecessary side effects. The authors designed a questionnaire which was answered by 58 from 165 RT departments in Italy. This represents a relevant proportion of the nation-wide radiation oncologists. The questionnaire is comprehensive and well-constructed. The issues are of great importance for radiation oncologists. The authors address several issues which are extensively discussed in recent literature, e.g. the radiotherapy dose prescription in distinct tumor stages and volumes, the value of PET/CT scans and the importance of gynecologic examinations. When considering the methodology, the authors could have additionally analyzed whether there are differences in the specific strategies in university hospitals or accredited cancer centers and public hospitals. However, the authors chose a broad approach and presented the results for all radiation oncologists of all institutions documented in Italy by AIRO. The authors provide a detailed discussion of recent literature and recent studies. It should be mentioned that similar surveys which reflect current treatment practice have rarely been published (e.g., Martin et al. 2020, reference number 15). The authors highlight the results on RT prescription doses, HIV screening, and involvement of gynecologists. When analyzing recent literature and recent guidelines on anal cancer, these issues are especially important and need further research. In summary, the authors uncover several aspects with considerable heterogeneity in the nation-wide strategies for anal cancer treatment. The presented data inform radiation oncologists on current treatment patterns. Additionally, these data can serve as a basis for further studies on patients with anal cancer, both in staging strategies and multimodal treatment strategies. Thus, an improvement of outcomes and a reduction of side effects can be expected on several levels. Lastly, the authors provide actual and appropriate references. The tables are presented clearly and transparently.Author Response
Reviewer 2.
The authors present a pattern of care report on the management of patients with squamous cell carcinoma of the anus. The questionnaire is comprehensive and well-constructed. The issues are of great importance for radiation oncologists. The authors provide a detailed discussion of recent literature and recent studies. There are no further remarks.
Thanks for your comments.
Second report (9.20):
The authors present a pattern of care report on the management of patients with squamous cell carcinoma of the anus. This topic is of great relevance for clinicians, since there is certainly a considerable heterogeneity in, e.g., staging examinations, radiotherapy dose prescription, and application of chemotherapy. Additionally, anal cancer is a rare tumor entity and radical treatment is often associated with high-grade toxicities. This poses a challenge to harmonize treatment strategies for optimal outcomes and to avoid unnecessary side effects. The authors designed a questionnaire which was answered by 58 from 165 RT departments in Italy. This represents a relevant proportion of the nation-wide radiation oncologists. The questionnaire is comprehensive and well-constructed. The issues are of great importance for radiation oncologists. The authors address several issues which are extensively discussed in recent literature, e.g. the radiotherapy dose prescription in distinct tumor stages and volumes, the value of PET/CT scans and the importance of gynecologic examinations.
When considering the methodology, the authors could have additionally analyzed whether there are differences in the specific strategies in university hospitals or accredited cancer centers and public hospitals. However, the authors chose a broad approach and presented the results for all radiation oncologists of all institutions documented in Italy by AIRO.
Thanks for your comment. This approach was chosen to be able to portray the situation in the whole country, in roder to sense eventual heterogeneity and call for tackling action for harmonisation.
The authors provide a detailed discussion of recent literature and recent studies. It should be mentioned that similar surveys which reflect current treatment practice have rarely been published (e.g., Martin et al. 2020, reference number 15). The authors highlight the results on RT prescription doses, HIV screening, and involvement of gynecologists. When analyzing recent literature and recent guidelines on anal cancer, these issues are especially important and need further research. In summary, the authors uncover several aspects with considerable heterogeneity in the nation-wide strategies for anal cancer treatment. The presented data inform radiation oncologists on current treatment patterns. Additionally, these data can serve as a basis for further studies on patients with anal cancer, both in staging strategies and multimodal treatment strategies. Thus, an improvement of outcomes and a reduction of side effects can be expected on several levels. Lastly, the authors provide actual and appropriate references. The tables are presented clearly and transparently.
Thanks for your comments.

Reviewer 3 Report
An interesting paper looking at difference in care but pretty much as I might expect
but no real plan or advice to either consolidate services or drive uniformity and quality of care
A missed opportunity for your patients
Author Response
Reviewer 3.
An interesting paper looking at difference in care but pretty much as I might expect but no real plan or advice to either consolidate services or drive uniformity and quality of care. A missed opportunity for your patients
Thanks for your comments. We are presently implementing initiatives to harmonise the managemnt of anal cancer patients in the country and to tackle the heterogeneity found in the present survey. We added few lines in the conclusion paragraph.
The Study Group for Gastrointestinal Tumors at AIRO is presently planning to set up a project based on the Delphi consensus methodology, to homogeneize diagnosis and treatment for anal cancer patients in Italy in order to mitigate the differences in patient management outlined by the present survey. After the generation of a consensus, we are also planning to register all anal cancer cases within a shared platform in the frame of a prospective observational cohort study.

Round 2
Reviewer 3 Report
Thanks for addressing my concern